# Urban System in Serbia—The Factor in the Planning of Balanced Regional Development

**Zora Živanović \*** , **Branka Tošić, Teodora Nikolić and Dragica Gatarić**

Faculty of Geography, University of Belgrade, Studentski Trg 3/3, 11000 Belgrade, Serbia
\* Correspondence: zoraz17@yahoo.com; Tel.: +381-113-570-946

**Abstract:** This paper analyzes the basic characteristics of Serbia's urban system after World War II. The term urban system is largely determined by the use and functioning of the space in which it exists. We used the methods common in urban geography, notably the Rank-Size Rule and the Law of the Primate City with the aim of identifying the basic regularities, as the first step in an in-depth study of an under-researched topic. The research seeks to contribute to explaining the causes of the previous and current situation in the national settlement network, as a prerequisite for planning the future organization of the settlement network. Our study, conducted in Serbia, finds polarization apparent in the prominent domination of the capital city in terms of population, and this is a key feature of Serbia's urban system. The current situation is the result of an intensive process of urbanization, but also of the establishment of new administrative boundaries after the disintegration of Yugoslavia. The study also seeks to suggest the most appropriate development model for Serbia's urban system that could help overcome the extreme population concentration in Belgrade and create a basis for organizing an optimal system of centers. Keeping in mind that uneven regional development is determined by the features of the urban system, polycentric development is a common model for overcoming extreme polarization on a global level.

**Keywords:** urban system; hierarchy; Rank-Size Rule; polycentricity; Serbia

## 1. Introduction

The paper seeks to determine the characteristics of Serbia's urban system (an urban system is a set of cities and towns in a territory interacting through the circulation of people, goods and information; it has the characteristics of a hierarchical system; for the development of the concept of urban system and the varying definitions see [1–5]), as one of the factors in the planning of balanced regional development; it deals with Serbia's national urban system. A national urban system is a set of interlinked towns and cities within the national territory, and a major change in the population size, economy, employment, service provision, etc. in one part of the system have repercussions in other places within the system [5,6]. Furthermore, the functioning of this set of urban entities may also be susceptible to government decisions or national policies [3]. Practice shows that the concept of urban development affects the overall development of an area. In other words, national urban systems are largely determined by the use and functioning of the space in which they exist, i.e., by the degree of its socioeconomic development and its social and political organization, and they may be closed or open to outside influences to various degrees [5].

Urban systems are subject to constant change, which is the result of spontaneous development or planned actions, and may also be caused by the changes of administrative boundaries in the area where they are located [7]. It is impossible to channel the future development of the network of urban settlements without knowing the specific urban system and its basic characteristics. This knowledge

is a prerequisite for any planned action towards organizing and planning an area in demographic, economic, social, infrastructural, or other terms.

The shift in the interpretation of urban systems that has occurred during the past decade relies on a broader perspective, according to which the global system consists of multiple, inter-connected networks. There is an equally strong interest in global cultural trends, political networks, media cities, and other forms of interconnectedness, including those associated with major infrastructure corridors [8].

The most striking features of Serbia's urban system include the pronounced primacy of the largest city—the capital—resulting in an uneven regional development. Among other things, the extreme concentration of the population, institutions, infrastructure, industry, traffic, and services in a single city or area has serious economic and environmental consequences, both for the capital and for the underpopulated interior of the country. It renders sustainable development impossible. According to Christopher et al. [9], in order to achieve environmental sustainability, urban consumption must match, or be below, the available natural resources, and the resulting pollutants must not exceed the environment's ability to provide resources. In Serbia, the extreme population concentration in Belgrade leads to illegal and unplanned construction, resulting in increased pollution (the urban heating and waste management infrastructure cannot develop in accordance with the population growth). Public transport cannot meet the needs of the growing population, and thus private cars are increasingly used, worsening pollution. On the other hand, public infrastructures in small towns become nonviable due to the declining population. Their poor operation and management also lead to increased pollution. No investment is provided for their development, which further encourages out-migration.

Although the consequences of uneven regional development are apparent, the research on the regional spatial structure in Serbia is still lacking. Furthermore, studies comparing the current situation with the period when Serbia was part of the Socialist Federal Republic of Yugoslavia are scarce, though it is apparent that the change of state borders and administrative boundaries affected the regional balance, and one would expect that such an important issue be subject to scholarly scrutiny. At the same time, there is a strong need for a regional balance and sustainable development and research in this direction is necessary.

Bearing in mind the state of the research, choosing the research method is a significant challenge. Furthermore, the available statistical data are incomplete, due to which the application of some complex and sophisticated methods is impossible. Therefore, we have decided to use a rather simple method, seeking to establish the basic facts that could serve as a starting point for a discussion on this topic. Our research seeks to test the Rank-Size Rule model in Serbia's national urban system. The results show that the Rank-Size Rule is a suitable method for describing the urban system in Serbia. This relatively simple and cheap method, relying on publicly available data provided by the national statistical office, captures the (otherwise apparent) misbalance of Serbia's urban system in terms of size through the striking deviation of the distribution of urban centers from the linear model. Through a descriptive historical analysis, the study seeks to explain the reasons for this misbalance, locating them primarily in the developments following the disintegration of Yugoslavia, but also in the fact that due to a political change Belgrade made a transition from the capital of a national urban system conforming to the Rank-Size Rule (SFRY) to the capital of a national urban system strongly deviating from it (Serbia).

Keeping in mind this last point, and the fact that an uneven regional development is rooted in the features of Serbia's urban system, and drawing on a significant body of research papers and policy documents, we believe that polycentric development is the most suitable model for overcoming the extreme polarization, or concentration. Although it is possible to find studies claiming that there are "too many empirical exceptions" to the Rank-Size Rule and "little substantive theory behind it" [10], recent studies [11] demonstrate that the Rank-Size Rule may be useful in elaborating the idea of a polycentric structure of urban systems at various territorial levels.

A balanced distribution of urban centers may enhance the overall development in the area where an urban system exists, i.e., this may have a positive effect on the development of underdeveloped

regions [12] and it may ensure sustainable development. In order to create a balanced and interconnected urban system, it is necessary to have multiple regional centers distributed throughout the country's territory. According to Faludi [13], there is a universal agreement that polycentrism is more likely to result in what is usually described as "territorial cohesion". Davoudi [14] confirms that "polycentricity now appears to be cropping up everywhere as an "ideal type" regional spatial structure." Numerous European strategic documents take a normative approach to the concept of polycentricity, as the chief guiding principle for achieving a regionally balanced development across the EU [15], setting the strengthening of polycentric development and innovation through networking between city regions and other cities as a goal [16]. In other words, they confirm that the principle of territorial cohesion through a balanced social and economic regional development and improved competitiveness are based on the model of polycentric development [17].

Polycentricity includes the rapid decentralization of economic activities; the increased mobility due to new transport technologies, the multiplicity of travel patterns, the fragmentation of spatial distribution of activities, the changes in household structure and lifestyle, and the existence of complex cross-commuting [14]. A major concern of the ESDP is "to reconcile the social and economic claims for spatial development with the area's ecological and cultural functions and hence contribute to a sustainable, and balanced territorial development" [15]. Document emphasizes that the development of the poor periphery is not to be achieved through outflows of resources from the affluent core [18]. It is clear that the ESDP takes a normative approach to the concept of polycentricity, advocating it as a preferred pattern of spatial structure and as a chief guiding principle for achieving regionally balanced development across the EU. Polycentricity includes the intra-urban scale or the level individual city, the inter-urban scale or the regional level, and interregional scale focusing on the level of EU as whole. At this latter scale, particular emphasis is placed on the conceptualization of polycentricity within the European spatial planning in general and the ESDP in particular.

## 2. Previous Theoretical Research

When explaining the development of an urban system, one usually highlights the significance of the initial advantages, which may vary in the different stages of the system's development [19]. In modern times, the importance of natural conditions and resources is relative because the location, expansion and development of a city are influenced by other factors (e.g., service industries, foreign trade, access to major corridors, etc.). The analyses of national urban systems show that, depending on the level of economic development, political situation, urban development tradition, situation in the international environment, the size of the country, etc., the Rank-Size Rule of a city may range from an irregular form with a high urban primacy index, through transitional forms, to fully regular ones. Within a particular time frame, the vertical dimension of centers corresponds to a spatial or horizontal dimension. An irregular Rank-Size Rule of cities usually corresponds to an irregular urban network with a prominent domination of the largest cities [20] (Christaller's central place theory is one of the first attempts to explain the spatial distribution centers and their functional hierarchical organization).

In their book Globalni grad (Global City), Grčić and Sluka [21] state: "The models of the development of urban networks are very diverse. At one pole, there is the model of a very balanced occupation of uninhabited space, whereas at the other, there is total superconcentration of the demographic potential in a single city-core."

A certain correlation between the size of the city (measured by the population size) and the number of cities was observed already in the early 20th century by the German geographer Auerbach (1913), when highly developed countries, including Germany, underwent the process of industrialization and urbanization, which resulted in the emergence of large cities. It was at that time that the regularity in the distribution of cities based on their size was observed and this regularity still persists. The population of German cities arranged in a descending order according to their size regularly declined in relation to the largest city. This regularity could be observed in most highly developed countries, but it was not possible to trace it in developing ones, as confirmed by research in many countries [22–25].

Auerbach's idea was later elaborated by many authors: Gibrat, Zipf, Berry, Stewart [26–29], and recently by Giffinger, Storper, Okumura, Overman, Rossi-Hansberg, Kim etc. [10,11,30–33], which has resulted in a model that explains the concept of the regular vertical distribution of urban centers in space. Zipf was the first to empirically demonstrate that the distribution of cities according to the population size in relation to their rank in a series is an approximately straight line when the two values are shown on a logarithmic scale; he thereby defined the Rank-Size-Rule [27].

The issue of the optimal and minimal sizes of the city also arose in the study of urban systems at an international level. They are associated with urban functions and their economic efficiency [34], or the ability to provide services in the city's gravitational area. Parallel to this, the concept of the optimal size of urban systems was developed [35], i.e., the optimal Rank-Size Rule of cities. The determination of the optimal size of urban systems inevitably leads to the question of the starting point for determining the optimum, i.e., the question of an urban system's Rank-Size Rule optimal for achieving certain objectives (e.g., the acceleration of economic growth, a more balanced interregional development, national integration, etc.). It turned out that a regular or optimal Rank-Size Rule may arise from an irregular Rank-Size Rule by increasing the level of development and urbanization.

Apart from the population size, analyses of urban systems based on the Rank-Size Rule used other indicators, such as the number of employees [36], or the influence of production activities [37], the size of the nonagricultural population, etc. Later studies [38,39] confirm Gibrat's empirical rule of growth, according to which cities at a similar level of development have a similar, proportional percentage of demographic growth [40]. The studies that seek to explain the city size distribution in several European countries [41] and China [42] confirm the administrative dependence on economic indicators. There is evidence that the growth of Italian cities depends on the quality of their urban location, i.e., the costs and benefits in the functioning of the city [43].

The studies conducted in many developed countries, e.g., Canada, in a later stage of development show a parallel growth of the size categories of cities in an urban system [44]. The examples of the southeastern region of the United States [45] and French cities [46] were used to test the correlation between the size of the city and the intensity of growth by calculating probability, assuming that Gibrat's Law [26] is applicable. Kim and Law [33] argue that political centralization in South America leads to a higher urban primacy and the distribution of city sizes is skewed toward large cities. According to Cheshire [47], during the 1980s, the process of re-centralization could be observed in many northern European cities. Regions that could attract workers and residents were able to improve living conditions and this usually led to re-centralization. The growth of large cities was accompanied with the growth of small towns, due to which the relations within the system have not changed significantly. Major changes in the systems arose with political and territorial changes, as evidenced by numerous examples from the history of urban development in many countries, including Serbia.

The stability of a national urban hierarchy in a temporal context has been confirmed by multiple studies. The Rank-Size Rule of cities is applied in urban systems in large countries with a long period of urban development and a greater political, economic, and spatial complexity, which may be readily confirmed by citing several examples:

- While exploring the urban hierarchy in the United States between 1970 and 1990, Henderson concluded that the size of the urban population in medium-sized cities (between 100,000 and 500,000 inhabitants) was stable, both in time and space, in different countries, and even in those where the economy was developing rapidly [48].
- Despite growth of the urban population of Brazil by 4–5% a year between 1950 and 1980, at the end of the observed period, the distribution of the urban population in the cities of different sizes was almost identical to that of 1950 [21].
- It took twenty years for Hiroshima and Nagasaki to regain the place in the urban hierarchy of Japan that they had had before World War II [49].
- The ratio of the population sizes of the largest and the second largest cities in France (Paris and Lyon) has been stable for two centuries [40].

Similarly to the Rank-Size Rule, a low urban primacy is, as a rule, found in large and industrialized countries with a long history of economic and urban development. This value may vary in different countries throughout history. Along with developing countries, a pronounced urban primacy is typical of the countries where the cities with a high urban primacy index were capitals of great empires, e.g., Vienna and Lisbon [40].

Empirical research conducted in several European countries has demonstrated that there is no direct correlation between the distribution types of cities according to the population size and the urbanization degree. Namely, a country may have a log-normal distribution of cities according to their demographic size and a low level of urbanization. Furthermore, the distribution of cities in a country may be marked by the domination of a single center accompanied with a high degree of urbanization, and vice versa [40]. As a rule, in the countries with a lower population concentration index, i.e., a lower degree of primacy of the largest (capital) city, it is possible to observe the conformity with the Rank-Size Rule of cities.

The pronounced polarity accompanied with the domination of, most commonly, the primary city, leads to a greater imbalance in the development of certain territorial units in the country. This phenomenon is more commonly present in countries with a lower development level, such as Serbia. The polarized development is discussed in numerous doctrines, which use a similar rationale but a different terminology to explain the phenomenon of polarized and, accordingly, unbalanced regional development. The concept of polarization is present in the work of many authors, e.g., Myrdal [50], in his theory dealing with the circular cumulative causation or an unbalanced regional development, or Friedmann [51], in the theory of the dichotomy between the center and the periphery. According to Friedmann, industrialization primarily led to a huge difference between the developed centers and the underdeveloped periphery. In his theory of growth poles, Perroux [52] argues that economic development is not achieved evenly and everywhere, but in certain points in space, such as the poles of growth, industrial hot spots, etc.

Apart from decentralization and the formation of a polycentric system of settlements, as a well-known model for overcoming the situation of extreme concentration, there are other ideas. As there is a problem of a high concentration of the population and economic activity in a small number of cities, it is suggested to apply the theory of economic growth, according to which the balance of the size and number of cities is determined both by agglomeration effects and by the level of productivity and communication costs [32].

## 3. Methods

The most common methods used in the analysis of urban systems belong to one of the two basic types: theoretical and empirical. The most frequently applied theoretical methods include the Central Place Theory, Gravitational Model, Graph Theory, Reilly's Model, Rank-Size Rule, the Law of the Primate City, etc. [35].

Due to the lack of appropriate empirical and qualitative methods or the inability to apply them, urban systems are often analyzed using quantitative methods. However, some authors believe that any quantification in regional science is useless and that it more often leads to wrong than to correct conclusions [53]. In accordance with the statement that the quantitative and qualitative characteristics of the analyzed phenomena are not opposed to each other, but are complementary, it is believed that quantification leaves behind the rest of the qualification and vice versa.

The Rank-Size Rule and the Law of the Primate City are often used in the analysis of the hierarchical or vertical dimensions of urban systems. The size of the city is usually defined by its population, though the application of the population size as the sole criterion in detailed analyses used in urban planning has many limitations. The same demographic size of settlements is not necessarily accompanied with the same economic strength, functional capacity, its importance for environmental development, etc. Along with the population size, studies related to education, as well as to the age, economic, and other structures of the population are particularly important. Nevertheless, the indisputable

fact that the population concentration implies the concentration of functions and capital, justifies the application of the population size as a sufficiently relevant indicator of an urban center's importance in a settlement network.

The regularity in the vertical distribution of cities may be explained using the Rank-Size-Rule, according to which the population of a city ($S_n$) in a series of cities ordered according to their size (measured by their population) corresponds to the population of the capital (largest) city ($S_1$) divided by the ordinal number of the observed city ($r_n{}^q$) in the series. This can be mathematically expressed using the formula [54],

$$S_n = S_1/r_n{}^q \tag{1}$$

Due to considerable differences in the sizes of cities, the Rank-Size Rule must be displayed on a logarithmic coordinate system, which is mathematically expressed as follows,

$$\log S_n = \log S_1 - q \log r_n \tag{2}$$

This rule corresponds to the linear functional correlation, where the ordinate (y) represents the rank and the abscissa (x) represents the population of the city, which is expressed as follows,

$$\log y = \log A - q \log x \tag{3}$$

A low value of the coefficient (q) indicates that the population is concentrated in a small number of cities, while a high coefficient reveals population dispersion among multiple cities. The value (A) represents the regression constant [30].

In many countries and especially in the developing ones, the primate (usually the capital) city stands out in terms of size. Already before World War II, this phenomenon was named the Law of the Primate City [55], and was explained by the exceptional political, economic, and even social character of the capital city [54]. The urban primacy index is the ratio of the number of inhabitants of the first ($G_1$) and the second largest city ($G_2$):

$$I_1 = G_1/G_2 \tag{4}$$

Urban primacy is high if the obtained ratio of the population sizes of the first and the second largest city is greater than 2, the ration of the second and the first city is greater than 1/2, the ratio of the third and the second city is greater than 2/3, etc.

In practice, the ratio of the population of the largest city and the sum of the populations of the next three largest cities ($G_2$, $G_3$, $G_4$) may also be used:

$$I_2 = G_1/(G_2 + G_3 + G_4) \tag{5}$$

## 4. How Is It That the Urban System Has Become Monocentric?

The polarization of the network of (urban) settlements in Serbia, as a crucial characteristic, is confirmed by the fact that, in 2011, almost one-third of Serbia's population was concentrated on only 5% of its territory, while the concentration of population in Belgrade accounted for 16.2% of the total population (as opposed to only 7% in 1948), or 27.5% of the urban population. The polarizing effects of urbanization, spatially manifested in the demographic and the economic and functional concentration, are the most apparent in the case of Belgrade. Already after World War II, Belgrade was the dominant development center, and its urban primacy was 6.21 in the territory of Serbia. In the following period, it was reduced to 5.03. The greatest population increase in Belgrade was recorded between 1953 and 1961, when the growth index was 137.5. Belgrade's demographic basis subsequently became increasingly huge and it could not keep on growing at the pace typical of the initial period, when the intensive process of industrialization and urbanization resulted in exceptionally high growth rates (30‰). After World War II, the primacy of Belgrade was less pronounced because the pace of its development was slower than that of the secondary city—Novi Sad.

As the development of regions is closely related to the characteristics of the regional centers within them and especially with the size and position of urban settlements in the network, i.e., in the hierarchical system of centers, the pronounced monocentricity, on one hand, and regional imbalance, on the other, are explained as mutually related and interdependent phenomena in the territory of Serbia.

Unbalanced regional development in Serbia can be explained in the following way, including relevant indicators. In its initial phase, the unbalanced regional development in Serbia was guided, as in most other countries, by the principles of polarization: intensive industrialization and urbanization, especially in the second half of the 20th century. The subsequent cumulative processes, accompanied with other factors, e.g., better infrastructure, concentration of skilled labor, positive externalities, etc., contributed to this trend, as well. In addition to regional inequalities, imbalance may also be observed at the local level, i.e., towns' influence areas, but, according to Glaeser et al. [56], it should "not be studied with the same analytical tools used to understand national inequality," especially in the case of non-European territories, which do not have much in common with Serbia.

The process of transition in Serbia in the late 20th century, accompanied with all the unfavorable circumstances and events in the country and external factors, led Serbia to become one of the most prominent examples of regional imbalance among European countries. First of all, the failure of massive privatization (the share of private-sector employees in the total labor force increased from 14.8% in 2000 to 25.3% in 2005 [57]), led to the collapse of industrial production (the number of workers in Serbian industry declined by more than half a million compared to 1990 [58]; the physical volume index of industrial production at the end of 2017 was two times lower (51%), i.e., it was at the same level as in 1972 [59], resulting in a large share of the unemployed population (over 25%, at the beginning of the 21st century [57]). Due to the collapse of industry in the 1990s, medium-sized towns lost their status as poles of development in their regional environment. Over the past several decades, this situation led to further migrations to larger towns, and especially to Belgrade and Novi Sad, as the secondary city, where employment opportunities were slightly more favorable. Even regardless of these circumstances, "larger cities lead to more amenity spillovers, which improve welfare and will attract the skilled" [60]. The migration process, which gave rise to many social problems, accompanied with prolonged economic crisis, led to economic destabilization, growing poverty in Serbia, and as a rule, the deepening of regional differences.

An insight into the economic asymmetry in Serbia and distinct regional differences may be gained from the following indicators.

- Serbia has the greatest inequality in the distribution of income among European countries; the Gini index for Serbia in 2009 was 28, and, in 2015, it was even more unfavorable—it rose to 38.6 [61].
- Twenty percent of the country's population with the highest incomes earns as much as nine times more than the poorest 20%, and this is the largest ratio in Europe [62].
- The disparities among 30 Serbian districts (NUTS 3) are great and they are illustrated by the following ratios; for the number of companies, 1:83; the number of employees, 1:64; the total income, 1:222; profit, 1:239; and loss, 1:91 [57].
- Complex economic analyses reveal extreme transitional regional imbalances at the district level; at the beginning of the transition, the extreme disparities were 7:1 (City of Belgrade vs. Toplica District—on the southeast of Serbia), in 2008 they increased to 16:1, but, in 2016, they were three times smaller, 5:1 [63].
- The ratio of the national income per capita of the poorest and richest municipalities in 2005 was 1:13.4, where the poorest municipality was Preševo, on the southeast of Serbia, and the richest was Apatin, on the northwest of Serbia [57].

There are also other reasons for the great regional imbalance in Serbia. A large part of Serbian territory lacks the potential and appropriate conditions for competitive development [64]. Less developed regions have either lacked the capacity or faced a delay in the process of accepting, developing and implementing high-technology products. Endogenous regional potentials were not

identified and, due to this, their development was not activated; furthermore, the operation of national sectoral institutions has been insufficiently aligned with the needs of regional development [12]. Besides, in Serbia, the regional policy has been of secondary interest, whereas the "focus of state instruments was dominantly placed on the process of transferring the planned economy into a market-oriented economy. The institutional framework of regional development in Serbia is asymmetric, nonfunctional and inefficient, not in the function of optimal resource reallocation" [59]. The degree of inequality has also been related to the inability of national tax and social policies to reduce inequalities in available income [61].

As far as governance is concerned, the main disadvantage of Serbia's current regional policy is the inadequate/underdeveloped network of regional development institutions that would provide human resources, policy, and technical support in defining and directing systemic and other measures towards a balanced regional development. Another reason for the rather poor outcomes of decentralization is the insufficient professional and technical capacity of the local administration to implement the reform measures [12]. In the process of planning and directing public aid, there is no efficient coordination among numerous institutions and entities involved in various aspects of regional development, such as economic development, rural development, infrastructure development, social development, local development, environmental protection, etc. Due to the asymmetrical regionalization, the regions within Central Serbia in particular have been left without the middle level of government, and, due to this, without regional mechanisms of financial assistance and coordination [64].

Therefore, despite the fact that the growth of the Belgrade agglomeration has been accompanied with the formation of counterbalancing development centers in the territory of Serbia, which relived the population pressure on Belgrade and contributed to alleviating the acute problems development unevenness problems by developing functions, the polarization in Serbia has reached large proportions.

At the same time, the drastically reduced growth rate of Belgrade's population was the result of a negative natural increase rate [65], both in Belgrade and in Serbia, as well as the population emptiness inside Serbia, which was no longer in a position to provide an increasing number of migrants ready to move to urban areas, and primarily to Belgrade. After the disintegration of Yugoslavia, the economic crisis, and political instability, the migratory movement has become more intense, but migrations have been directed towards developed countries. Emigration and a low natural increase in Serbia result in the annual loss of 30,000–40,000 inhabitants [66].

Accordingly, the primary effects of the population concentration in Belgrade's metropolitan area have been a result of an intensive in-migration process (since 1991, the natural increase has been negative) from throughout the national territory. In the entire body of the population that out-migrated from the municipal territory during the period of intensive village-to-city migrations, the proportion of those who moved to Belgrade was greater than the share of those who chose a municipal center [67] or the nearest urban settlement as their destination. This emphasizes the role of Belgrade as the most significant demographic pole in the territory of Serbia.

The reasons for Belgrade's extreme primacy in the urban system of Serbia are also associated with the change of the state borders in recent history. Throughout history, the existence of Serbia as a state has almost exclusively been associated with Belgrade as its capital. In this respect, its extremely important role within national borders, though guaranteed, varied greatly in different periods or different contexts in which the city was the capital.

As part of the SFR Yugoslavia, the most developed national centers apart from Belgrade and important development leaders in their environments were Zagreb, Skopje, Sarajevo, Ljubljana, and Podgorica, as the capital cities of the republics (federal units) that were part of Yugoslavia. Their demographic and economic power increased with varying intensity.

According to the 1921 Census, Zagreb, as the second largest city in the former Yugoslavia, had only 3000 inhabitants fewer than Belgrade, whereas, in 1948, this difference reached almost 120,000 inhabitants. After World War II, urban centers witnessed significant economic and demographic prosperity. It may be provisionally claimed that the rate at which the population of Zagreb, as the

second demographically strongest city in SFR Yugoslavia, was growing was half the value of this indicator for Belgrade. In 1991, Belgrade was 1.5 times higher than Zagreb, but this ratio had been substantially constant in the period after World War II (values $I_1$, Table 1). Belgrade's primacy was not prominent even when its demographic size was compared to the sum of the populations of the next three largest cities (value $I_2$). At the end of the observed period, Zagreb was 1.75 times larger than Skopje, the third largest city in the former Yugoslavia, etc. This means that the urban system of the former Yugoslavia conformed to the Rank-Size Rule, i.e., the log-normal distribution of cities according to their demographic size, without the prominent primacy of the capital city.

**Table 1.** The population of the capitals (in thousands) of the republics of the former SFR Yugoslavia and the value of the urban primacy index ($I_1$ i $I_2$).

| Population in the Capitals | 1948 | 1953 | 1961 | 1971 | 1981 | 1991 |
|---|---|---|---|---|---|---|
| Belgrade | 397.7 | 477.9 | 657.3 | 899.0 | 1087.8 | 1168.4 |
| Zagreb | 279.6 | 350.6 | 485.0 | 629.9 | 723.1 | 777.8 |
| Skopje | 87.6 | 118.7 | 180.9 | 308.0 | 408.1 | 444.3 |
| Sarajevo | 113.8 | 135.7 | 179.8 | 271.0 | 319.0 | 407.0 |
| Ljubljana | 115.1 | 138.5 | 157.4 | 208.1 | 224.8 | 272.0 |
| Podgorica | 10.3 | 16.4 | 27.0 | 55.0 | 96.1 | 117.9 |
| $I_1$ Primacy Index | 1.42 | 1.36 | 1.36 | 1.43 | 1.50 | 1.50 |
| $I_2$ Primacy Index | 0.83 | 0.79 | 0.78 | 0.74 | 0.75 | 0.72 |

Note: The table relies on the data available in [68].

The development policy of Slovenia, which favored a decentralized system, resulted in a more balanced development of urban centers, due to which, Ljubljana, as the capital city, did not significantly stand out in the urban network of the former Yugoslavia, and even today, it does not stand out in the current urban network in Slovenia.

Not so long ago, the ranking of the Belgrade region based on the achieved level of social and economic development, as well as the overall prosperity and vitality, was considerably higher than that of the regions of the capital cities in former socialist East European countries: Budapest, Bucharest, Sofia, and others in the immediate neighborhood. After a decade-long crisis that befell Serbia after the disintegration of Yugoslavia, i.e., after the secession of its republics, this ranking has changed to the detriment of Belgrade, as substantially illustrated by a sole example: the turnover in air transport in Belgrade in the first decade of the 21st century was four to five times smaller than that of Bucharest and Budapest [69], though it had been considerably greater before the 1990s.

## 5. Results

Serbia's current urban system is marked by a prominent dominance of the primate city and a pronouncedly small number of large cities, as well as the lack of urban settlements with a demographic size ranging between 300,000 to 1,000,000 inhabitants. The current situation leads to the disproportion, incoherence and asymmetry in the system of cities, the hierarchy of which does not conform to the Rank-Size Rule. The "missing" cities would be the bearers of macroregional functions and would contribute to the internal balance in the development of Serbia. The "missing cities" would also help ensure the sustainability of the national urban system. Namely, the viability of public transport, centralized urban heating systems, and water and waste management facilities increases with urban size [70].

The context of the national urban system is defined by 164 urban settlements: 112 in Central Serbia and 52 in Vojvodina (as after 1981 Census did not cover the inhabitants of Kosovo and Metohija, the present study will highlight only the basic demographic characteristics of urban areas in Central Serbia and Vojvodina; for the same reason, the analysis of data for Serbia as a whole does not cover this province). The plausibility of the distinction between urban and non-urban settlements is a disputable

matter. Some experts believe that it is necessary to redefine the way of determining urban settlements, which would highlight an even lower level of urbanization and prominent polarization. In Serbia, after World War II, the official statistical service has used two criteria for the classification of settlements, i.e., the distinction between urban and non-urban settlements. The first criterion is administrative. According to it, settlements are officially proclaimed as urban by means of legal instruments. The other criterion is demographic statistical, obtained by combining two characteristics: the size (population) and the percentage share of the nonagricultural population [71]. Recent studies related to Serbia propose the distinction between urban and rural settlements based on multivariate analysis—PCA, FA, and CA [72].

Although the official statistics in Serbia do not recognize the categorization of urban settlements according to their size, we may conditionally accept the division into four categories (Table 2 and Figure 1):

- large cities (4), with more than 100,000 inhabitants (Belgrade, Novi Sad, Niš, and Kragujevac),
- medium-sized towns (36), with a population between 20,000 and 100,000,
- small towns (32), with a population between 10,000 and 20,000, and
- very small towns (92) with less than 10,000 inhabitants.

**Table 2.** The population size and the number of urban settlements in Serbia in 2011.

| Category | Number of Urban Settlements | Population | Share in the Population of Urban Settlements (%) | Share in the Total Population (%) |
|---|---|---|---|---|
| >1,000,000 | 1 | 1,166,763 | 27.45 | 16.23 |
| 300,001–1,000,000 | | | | |
| 100,001–300,000 | 3 | 565,797 | 13.31 | 7.87 |
| 50,001–100,000 | 13 | 858,507 | 20.20 | 11.95 |
| 20,001–50,000 | 23 | 730,582 | 17.19 | 10.17 |
| 10,001–20,000 | 32 | 485,547 | 11.42 | 6.76 |
| 5001–10,000 | 42 | 309,271 | 7.28 | 4.30 |
| <5000 | 50 | 134,563 | 3.17 | 1.87 |
| Total | 164 | 4,251,030 | | 59.15 |

Note: The table relies on the data available in [66].

Regardless of the small number of large cities in Serbia, the effects of the pronounced spatial and demographic polarization are apparent. In terms of numbers, small urban settlements prevail: settlements with up to 10,000 inhabitants account for more than a half (56%) of all settlements. As far as the smallest centers are concerned, eight of them have less than 1000 inhabitants, and these are mostly mountain or spa tourist resorts, as well as urban settlement in the Belgrade region. Apart from the largest cities, medium-sized towns, with populations between 20,000 and 100,000, are the most important in the spatial and functional organization, as they serve as regional centers of administrative territorial units in Serbia.

An intensive urbanization in Serbia was undertaken as late as the second half of the 20th century. In terms of economic structure, this area was entirely agricultural in character before the 1940s, whereas, in terms of settlement structure, this was a rural environment. According to the 1953 Census, approximately one-fifth of the total population (22.5%) lived in urban areas and approximately two-thirds of the active population (67%) was agricultural population [73]. As the degree of urbanity increased to 59.2% by 2011 [66], the share of the active agricultural population decreased to about 15% [74].

The urban centers mentioned in this study are mainly medium-sized towns located in river valleys or along the major traffic corridors in Serbia. Their status is not related to the vicinity of large cities and major development areas, as their development is determined by other factors. The development

of major industrial facilities after World War II and an intensive urbanization and deruralization were
the crucial factors of urban growth.

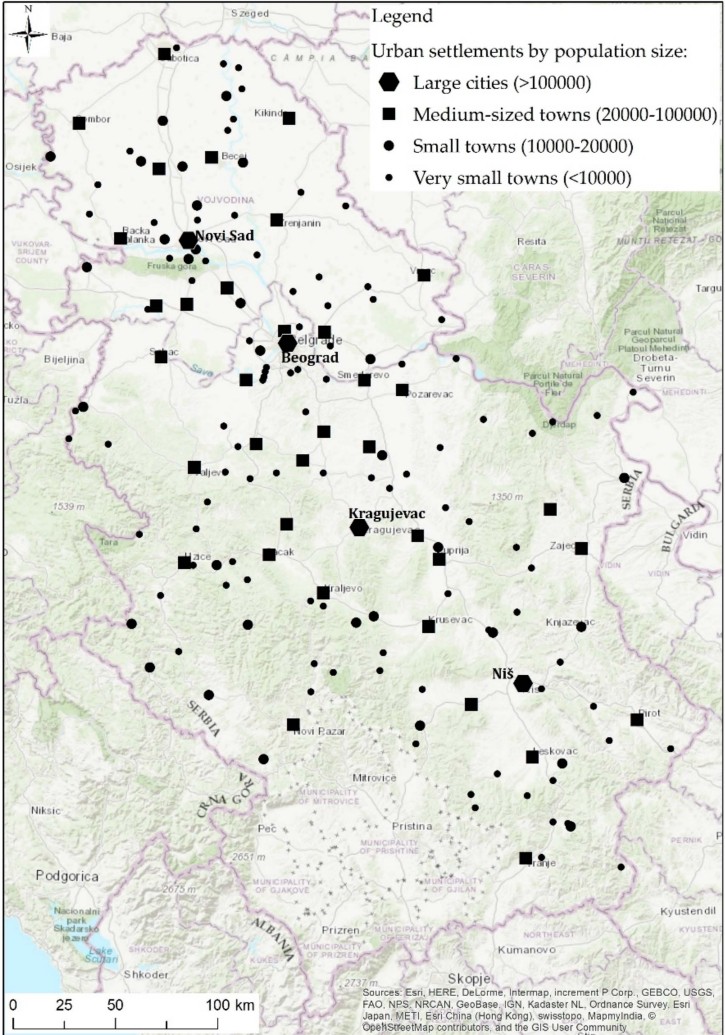

**Figure 1.** Urban settlements in Serbia, population in 2011. Note: The figure relies on the data available
in [66].

These processes were politically induced, and local potentials had a negligible role in urban
growth. Namely, political decisions guided the establishment of factories, seeking to provide every
town with "something", not relying on an analysis of available resources in the immediate environment.
This was an unsustainable basis for production in the long-term. Having left the virtually emptied
rural areas, people found jobs in large factories, the majority of which would be closed in the 1990s,
following an unsuccessful privatization. Regardless of the successful growth of medium-sized towns,
the tendency towards settling in major urban centers, primarily Belgrade, was present throughout the
period following World War II [75].

*The (Non-)Conformity of Serbia's Urban System with the Rank-Size Rule*

Based on the census statistics for the period between following World War II, where the beginning
and ending years are 1948 and 2011, we tested the validity of the Rank-Size Rule parameters for
describing the size distribution of cities according to a logarithmic distribution (base *e*, Figure 2), with
appropriate linear models:

1948: ln (rank) = 9.3412 − 0.6388 ln (popul.)

2011: ln (rank) = 9.7699 − 0.6202 ln (popul.)

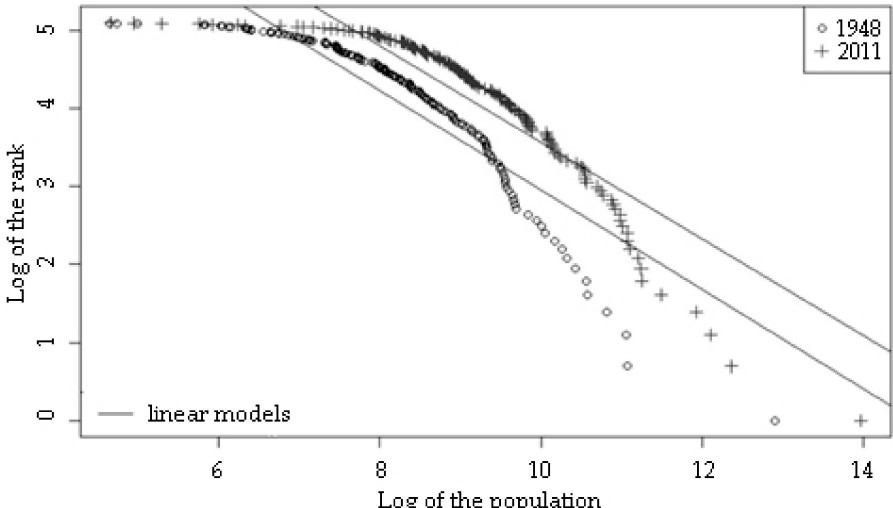

**Figure 2.** Logarithmic distribution of city sizes in Serbia between 1948 and 2011. Note: The figure relies on the data available in [66].

The small difference of coefficients (−0.6388 < −0.6202) reveals a slight increase in the nonuniformity of the urban population after World War II.

The frequency of cities according to the logarithmic values of population shows that the majority of urban settlements fall between $e^8 = 2980$ inhabitants and $e^{10} = 22{,}026$ inhabitants (Figure 3).

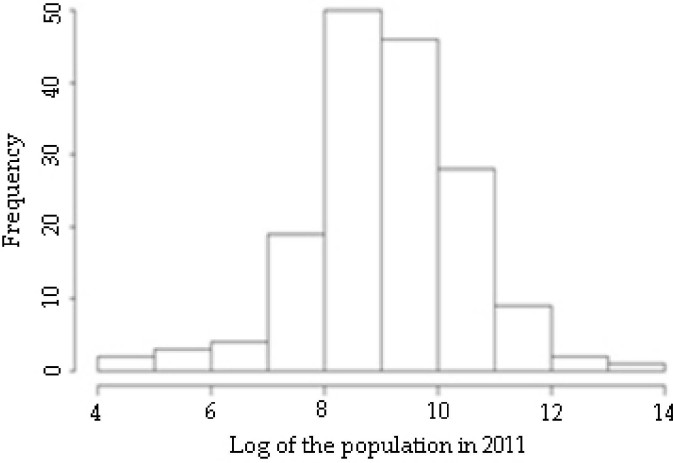

**Figure 3.** Frequency of cities according to the logarithmic values of population. Note: The figure relies on the data available in [66].

In Serbia, the ratio of the population of Belgrade, as the capital and the largest city, and Novi Sad, as the second largest city, is clear-cut, and is one of the greatest in Europe: the primacy index reached the value higher than 5 in 2011. In the observed period (1948–2011), the values of this index slightly decreased (Table 3, value $I_1$).

The scale of Belgrade's domination over other cities in Serbia is also apparent from the index that represents the ratio of the capital's population and the sum of the populations of the next three largest cities. The index value ($I_2$) was greater than 2 throughout the observed period and, judged by the rules of this method, it was significantly increased, regardless of the fact that the value decreased during the observed period.

According to Census data, the average value of the urban population (*Avg*) grew after World War II, from 9520 inhabitants in 1948 to 25,909 inhabitants in 2011, but the period after 1991 was marked by stagnation. Standard deviation values (SD) tripled during the observed period (Table 3).

**Table 3.** Values of the urban primacy index $I_1$ i $I_2$, the average city size (*Avg*), and the standard deviation of the city size (SD) in Serbia after World War II.

| Population in the Cities | 1948 | 1953 | 1961 | 1971 | 1981 | 1991 | 2002 | 2011 |
|---|---|---|---|---|---|---|---|---|
| Belgrade | 397,678 | 477,942 | 657,302 | 899,004 | 1,087,804 | 1,168,409 | 1,119,523 | 1,166,763 |
| Novi Sad | 64,041 | 70,769 | 95,192 | 134,160 | 163,773 | 173,186 | 191,656 | 231,798 |
| Niš | 47,296 | 56,589 | 78,712 | 124,264 | 157,326 | 175,649 | 175,631 | 183,164 |
| Kragujevac | 39,324 | 48,702 | 63,347 | 92,985 | 129,017 | 147,305 | 146,373 | 150,835 |
| $I_1$ primacy index | 6.21 | 6.75 | 6.91 | 6.70 | 6.64 | 6.75 | 5.84 | 5.03 |
| $I_2$ primacy index | 2.64 | 2.71 | 2.77 | 2.56 | 2.42 | 2.35 | 2.18 | 2.06 |
| Avg | 9520 | 10,971 | 14,205 | 19,109 | 23,495 | 25,555 | 25,575 | 25,909 |
| SD | 32,130 | 38,386 | 52,566 | 71,951 | 87,265 | 93,743 | 90,297 | 94,452 |

Note: The table relies on the data available in [66].

Serbia has too few cities with a population larger than 100,000 inhabitants (4), and this number demonstrates the deviation from the theoretical assumption, according to which there should be 15 such cities in order to obtain a Zipf's coefficient smaller than one. The differences in the population size between the second, third and fourth cities are even smaller than the values derived from the Rank-Size Rule. However, the greatest deviation is observed in the population sizes of the first and the second largest city, Belgrade and Novi Sad. Beginning with the fifth largest town, to the end of the series, the deviations from the Rank-Size-Rule are insignificant [40]. This means that medium-sized and small towns conform to the Rank-Size Rule.

The interdependence of the logarithmic population size of urban settlements in 2011 and the index of population growth (1948–2011) were analyzed in accordance with the rules set out in Gibrat's Law. The interdependence shows a low correlation (0.13869), which means that there is a low connection with the rule that large cities have a higher population growth (Figure 4).

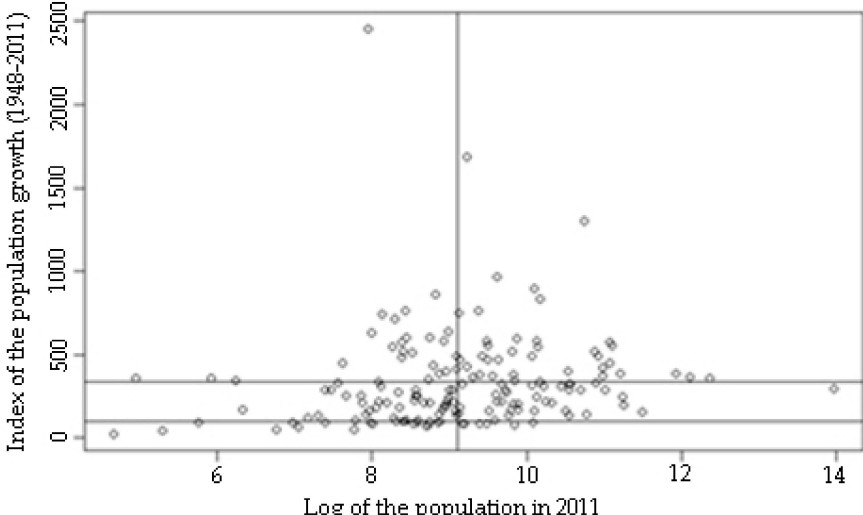

**Figure 4.** Logarithmic values of the population size of urban settlements (*X*) and the index of population change after World War II (*Y*). Note: The figure relies on the data available in [66].

$$cor(X, Y) = \frac{\sum_{i=1}^{n}[[x_i - \bar{x}] * [y_i - \bar{y}]]}{\sqrt{\sum_{i=1}^{n}(x_i - \bar{x})^2 * \sum_{i=1}^{n}(y_i - \bar{y})^2}} = 0.13869 \qquad (6)$$

where: $n$: the number of cities; $x_i$: log population of the city; $y_i$: index of the population growth, $\bar{x}$: average log city population, and $\bar{y}$: average index of the population growth.

The average value of the logarithmic population of urban settlements is 9.10791, and the average population growth in the observed period is 335 index points, as indicated by boundary lines (Figure 4). The largest number of cities is grouped around the mentioned average values. As far as the intensity of population change is concerned, three urban settlements prominently stand out. These are the mountain tourism center at Zlatibor; the municipal center Tutin, with a majority Muslim population with a high birth rate; and Borča, a suburban center near Belgrade. The population decreased in 15% of cities (the third boundary line in Figure 4 indicates the index of population growth with the value of 100).

If the number of cities according to their position in relation to both average values is analyzed, it can be concluded that the largest share of cities—more than one-third—had below-average growth and a below-average log population. In only 14% of the cities, the average growth was above average, but the log population size was below average (Table 4).

**Table 4.** The number of cities in relation to the average values of the log population size and the index of population growth.

| | Gibrat's Law Distribution | | |
|---|---|---|---|
| Average growth index of cities 1948–2011 (335 index points) | Average log population value of cities (9.10791) | | |
| | No. of cities > average | No. of cities < average | Total |
| No. of cities > average | 39 | 23 | 62 |
| No. of cities < average | 46 | 56 | 102 |
| Total | 85 | 79 | 164 |

## 6. Discussion

The second half of the 20th century was the period of the most significant expansion of the boundaries of the Belgrade region or the zone of its direct influence. An extremely rapid development of Belgrade led to imbalance between Belgrade and other urban centers, especially those in the territory of Serbia. The high value of its primacy index, which is today a characteristic of Serbia, dates from this period. In other words, the domination of Belgrade was considerably less pronounced in the former Yugoslavia than it is today in the territory of Serbia.

Along with the constantly intensive population concentration in Belgrade after World War II, which was more pronounced than the same trend in all other cities of the former Yugoslavia, it was Yugoslavia's disintegration (the disintegration of SFR Yugoslavia also led to the fragmentation of its urban system into five more or less incoherent and incompatible national urban systems), after which Belgrade became the capital of only one republic, that further strengthened its position against the under-developed network of macroregional centers in Serbia. Niš and Kragujevac, in Central Serbia, and Novi Sad (and until recently Subotica), in Vojvodina, as the only urban settlements with more than 100,000 inhabitants in the last decade of the 20th century, were given a chance for a more intensive development as the leading development centers within the national territory, apart from Belgrade. However, due to war destruction and the collapse of the economy in Serbia, as well as the fairly emptied rural hinterland of these cities, it has not been possible to intensify their growth and development, in terms of population concentration and, consequently, the availability of other facilities. At the same time, Belgrade has retained the prominent primacy of its position.

Due to the lack of an appropriate regional development policy, conditions have been created for a constantly intensive population concentration in Belgrade and the consequent agglomeration of activities in this city. This has resulted in the establishment of an uneven national urban system and the emergence of an unbalanced regional development (the ratio of the GDP per capita in the richest and the poorest regions in Serbia, the NUTS 3 level, is 15.3:1 [57]. The constant concentration increase leads

to the growth of Belgrade's power in the wider region of the Western Balkans and even to the overall development of Serbia. Nevertheless, in poorly developed countries like Serbia, further polarization is a threat to the development of the country's interior. To mitigate this and to ensure a balanced regional development, polycentric development is suggested as a necessary planning measure. The apparent difference in the demographic strength of Belgrade as compared to other macroregional centers highlights the need to organize the national territory so as to enable the decentralization or rather demetropolization or "de-Belgradization" of Serbia.

## 7. Conclusions

The unevenness of Serbia's development has primarily been caused by the distinct metropolization and polarization of space. The following facts determine the characteristics and the process of the development of Serbia's urban system.

- The urban system is marked by Belgrade, i.e., the capital city's prominent domination in terms of size (the urban primacy index has the value of 5.03 with respect to the second largest center, and Novi Sad, its value is more than 2 with respect to the sum of the populations of the next three largest cities—Novi Sad, Niš, and Kragujevac).
- The values of the urban primacy index have slightly decreased after the 1970s, when they reached the maximum.
- The Serbian urban system lacks cities with a population between 300,000 and 1,000,000, and this significantly contributes to Belgrade's prominent primacy and the lack of conformity with the Rank-Size Rule of cities.
- There are a small number of large cities, with more than 100,000 inhabitants—only four.
- More than one-half (56%) of the towns in Serbia's urban system are the smallest ones, with less than 10,000 inhabitants.
- Most urban settlements show values close to the average value as regards their size and population growth.
- The intensity of urban settlements' growth depends very little on their size.
- Under the former Yugoslavia, the urban system of the country conformed to the Rank-Size Rule (Belgrade's urban primacy index was only 1.5 in relation to Zagreb), which confirms the principle that the regularity in the size hierarchy of urban systems is directly related to the size of the territory.
- With the disintegration of the former Yugoslavia the degree of polarization, i.e., the size primacy of the capital in the territory of Serbia has significantly increased.

The reaffirmation of the principle of polycentricity, which basically implies an equal representation of all sizes of cities, confirms the topicality of this concept as a desirable development pattern, pursued both in Europe and in the world, and the basis of the current European principles of regional planning. The measures aimed at overcoming the consequences of the unbalanced regional development proposed by theorists of the mid-20th century seek to establish a regular hierarchy of urban centers, which implies the establishment of a polycentric system centers. Serbia's commitment to the development of this concept is highlighted in the Spatial Plan for the Republic of Serbia [76]. It is reaffirmed in the latest adopted document Spatial Plan of the Republic of Serbia [69]. However, Serbia's economic situation is a major limiting factor in the implementation of these goals.

Although we are aware of the indisputable domination of Belgrade in the settlement network of Serbia, especially in its central part, we believe that a group of medium-sized cities and macroregional centers may act as a pivot for (establishing) a (homogeneous) national urban system. Positive effects of polycentricity, both at the inter-regional [77] and intraregional levels [78] have been confirmed through studies relating to separate and distinct cities or smaller settlements that interact with each other to a significant extent. Unfortunately, the usability of data related to medium-sized towns in Serbia

for this purpose is very limited. The insight into previous research at the global level suggests that small towns have received insignificant attention, although in some parts of the world, e.g., Africa, small and medium-sized urban settlements are growing most rapidly [79]. Most medium-sized towns in Serbia were exposed to depopulation between 2002 and 2011. The active population involved in production stagnated or declined, whereas the data indicating the growth of the services sector were not realistic indicators of development but rather of the rise of shadow economy trade (without a legal foundation). Sporadic development of small and medium enterprises involved in low-accumulation activities merely mitigated the high unemployment rate (over 20% in Serbia) in medium-sized towns.

Despite these facts, we believe that it is plausible to treat these towns as the relevant bearers of development processes, as their presence in the settlement network suggests that its consolidation is possible. Investment, accompanied with state incentives could encourage population influx in medium-sized and small towns, where the increasing significance of "economies of agglomeration" and "clusters of activity" [80–82] in the distribution of employment and population could increase the competitiveness of these towns and, consequently, mitigate the existing regional disparities. Their growth would also make the development of local infrastructure and services viable. This approach would result in an increased consistency of the settlement network and the overall sustainability of the urban system.

**Author Contributions:** Conceptualization: Z.Ž. and B.T.; Methodology: Z.Ž. and B.T.; Validation: B.T.; Formal Analysis: Z.Ž., T.N., and D.G.; Investigation: B.T., Z.Ž., and T.N.; Data Curation: Z.Ž.; Writing—Original Draft Preparation: D.G.; Visualization: Z.Ž.; Supervision: B.T. All authors read and approved the final manuscript.

**Funding:** This research was funded by the Ministry of Education, Science and Technological Development of the Republic of Serbia within the projects nos. 176017, 47006, and III 47014.

**Conflicts of Interest:** The authors declare no conflicts of interest.

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
