# Peer review of "Urban System in Serbia—The Factor in the Planning of Balanced Regional Development"

_sustainability, doi:10.3390/su11154168_

Reviewer 1 Report

The subject of the paper is interesting and relevant as well as the results. The authors have made efforts to review and summarize the research area, which will provide useful information for future researchers and practitioners in the field of urban development. I found the paper to be overall well written. The presented text meets the formal criteria of scientific studies: the empirical analysis is based on a theory and the findings are based on the conducted analysis, but in my opinion it needs a little effort before acceptance.
Therefore, I recommend that a minor revision is warranted. I explain my concerns in more detail below.
1. My main comment concerns the description of statistical methods used in the paper. I have doubts about the nedd to present in the main text a quite simple and popular formula of the Pearson coefficient in its current form. In my opinion, this information can also be shown in the main text without the presenting a very well-known formula of this coefficient, especially in the scientific paper.
2. The figures presented in the paper have to be shown in more clear form. In current form they are illegible.

Author Response

Thank you for suggestion.

We are not sure if you think about coefficient line 324 ? If yes, we will remove it.

The figures presented in the paper are improved.

Reviewer 2 Report

The paper is very well structured and written. The research methodology is appropriate. The findings are of interest to researchers and policymakers. 

Author Response

Thank you for suggestion.

We are very grateful for recommending article for publication.

Reviewer 3 Report

See review report attached.

Author Response

Thank you for your review. We have read it thoroughly and we will try to respond to your comments, though we must say that they are rather general, insufficiently specific and not always clear, partly due to grammar and syntax. Due to this, our responses may also be rather general.

The shape of a research paper is determined by many factors. It is not the same if the authors pick a topic embedded in an elaborate network of previous research, involving a huge body of data and a plethora of methods already tested in previous studies, or if they choose a topic from a rather unexplored and unstructured complex of problems. Unfortunately, we happen to live in a country where there are many research problems of the latter kind and we happen to receive public funding to explore them. For many decades, local research was rather limited in scope and repetitive in terms of topics and methods, due to which the authors who seek to discuss a seemingly local topic in a broader context and present their findings to an international audience face many difficulties, such as insufficiently granular and incomplete public statistics (which is often difficult to compare with, e.g. statistical data collected in the EU due to different data collection methodologies), or the lack of previous research to draw upon. These limitations inevitably shape the research and are reflected in research papers, while the indicators are often limited to what is available (though we are well aware what would be optimal).

An early version of the present manuscript was considerably different – much more focused on the empirical study and, in our opinion, better articulated. However, our international colleagues who were kind to read it and comment found it difficult to understand the idea and the purpose of the study without details concerning the historical and social background. On the other hand, our local colleagues found those sections superfluous, as they were familiar with the context. Unfortunately, it was impossible to solve the problem by referring to the relevant literature because it was either nonexistent or inadequate. That is why we had to include the extensive ‘descriptive’ sections, although they might affect the articulation of the study.

Our study does not seek to explain everything. It is indeed limited in scope (it uses a very simple method to measure the apparent and extreme disproportion of an urban system based on available data and suggests a possible solution, with planning implications, towards achieving greater sustainability) and its main ambition is to stir discussion on an underpublished topic (at least in a regional context) and encourage researchers to seek for (alternative) answers to the issue of mitigating the extremely unbalanced regional development using different methods. Bearing in mind the current state of research and the relationship between the local planning theory and practice, providing an actionable policy or a definitive explanation at this stage of research would be overambitious.

We also provide response to your minor remarks. Based on the minor remarks, corrections in the text have been made (red text) and we hereby provide additional explanations:

In the introduction L30, what sense is given to “optimal”? As complex systems, urban and territorial systems may optimize many objectives, more or less contradictory.

In this context, optimal means the most suitable, the most efficient, or the most practicable. Keeping in mind the local context in Serbia and relying on a significant body of research papers and policy documents, we believe that polycentric development and related planning policies would be the most efficient in overcoming extreme polarization, or concentration (in terms of population, resources, infrastructure), as the uneven regional development is rooted in the features of the urban system.

The positioning regarding network literature is not clear, and complexity approaches to cities are not new indeed. Providing only (Brenner, 2016) for such a positioning is not satisfying.

Please check the chapter Previous Theoretical Research.

The discussion on scales and polycentricity is relevant and useful, but should be done with higher caution. Indeed, definition and measures of polycentricity are not a subject of consensus, and these furthermore depend on scale. Polycentricity in the sense of mega-city regions (Hall and Pain, 2006) is rather dierent from a macroscopic polycentricity at the scale of the system of cities (let say a low urban hierarchy). The citation train L55 (which is furthermore not recommended) makes a confusion across dierent scales and underlying concepts.

Corrected.

There is very few discussion on the definition of cities - but these highly influence scaling law patterns as shown by Cottineau et al. (2017). The choice made in the paper and its influence should be more thoroughly discussed.

In the present study we decided to stick to the definitions used by the national statistical office. It would certainly be interesting to conduct an analysis similar to that in Cottineau et al. (2017) in Serbia but this is beyond the scope of this study. However, it could be very useful in the further stages of our research.

What are “vertical” and “horizontal dimension of centres” (L82) is unclear.

The vertical dimension of centres refers to the hierarchy of cities according to their demographic size, whereas the horizontal dimension refers to the spatial distribution of cities. An irregular Rank-Size Rule with a high urban primacy index usually corresponds to an irregular urban network with the largest cities on its periphery (Vresk, M. 1984).

Citation L85 is too long, should be synthesized.

Corrected in the text.

In Table 1, how population classes are chosen should be made explicit. Is it an a priori classification or does it result from preliminary data analysis

Population classes are chosen as a result of preliminary data analysis. They are also generally accepted in the local literature.

End of page 6 and page 7, it is not clear if these are results actually obtained through data analysis, or a literature review or discussion of previous historical knowledge on the Serbian urban system.

These results are obtained through data analysis and through literature review (see the sources cited in the list of references).

Footnote 2, page 6, partly deals with the issue of the definition of cities. This would be relevant to be more developed and included in main text.

Corrected in the text.

Footnote 3, page 6 : to what extent removing these regions could bias the analysis, for exemple through missing urban interactions or missing medium-sized cities ?

It is impossible to provide a precise answer to this question, as the relevant data are missing for a period spanning almost 40 years.

Page 8, on fitting the rank size rule with a linear regression : adjustments are poor, adjusted R-squared should be provided. In that context, the slight dierence in the coecients estimated in time may be not statistically significant. Furthermore, a more elaborated investigation could be done given the shape of rank-size profiles, for example a piecewise linear fitting (superposition of two or more rank-size systems), a better estimator for the power law taking into account minimal cut-o (see (Clauset et al., 2009)), or even more elaborated geographical interaction models (for example (Favaro and Pumain, 2011)).

We believe that the method we applied in the present study is adequate for the main purpose of the study. However, the suggested methods could yield useful information and we will consider testing them in the further stages of our researc.

The discussion on economic disparities between L384 and L396 deals with socio-economic stylized facts. How can these be empirically related to the empirical results of the paper which are on population only, other than through untested hypothesis?

The purpose of citing the mentioned information is to offer a broader context to the readers. The mentioned lines are not directly related to the empirical results of the study, but they are illustrative of the local context.

At the end of the discussion, when discussing the former conforming of Yugoslavia to the rank-size rule and how its disparition lead Serbian urban system in its current state, more emphasis should be given to the question of system definition, and possible superposition of subsystems when estimating scaling laws, as this underlying issue is crucial and implicit throughout the paper.

Please note that the wording of the question is unclear and we are not sure whether we have understood the question correctly.

The issue remains implicit because an attempt to explain it in greater detail would lead us into another extensive descriptive analysis of the historical context, the one preceding the formation of the First Yugoslavia. Yugoslavia’s urban system was a combination of multiple urban systems rooted in different traditions and shaped by the planning policies of different regional powers (as some parts of Yugoslavia had previously been part of other political entities). The more balanced regional development under Yugoslavia was only partly due to an intended system definition and was largely determined by the historical and political background.

The last sentence of the conclusion is more an assumption than a conclusion of the paper. Either empirical research direction to support it should be provided, or this should be made more explicit.

Corrected in the text.

Reproducibility.

The paper does not provide data nor code used for the analysis and is thus not reproducible. The authors should provide the dataset used and the code for a higher scientific soundness.

Although the journal’s policy requires authors to to make research data openly available, only a limited number of articles published in Sustainability have supplementary data. Moreover, during the submission process we were not prompted to upload data. The journal’s data policy is very general: it does not say whether the reviewers are obliged to treat this kind of information as confidential material, nor does it define precisely the procedures in case we wished to deposit data in a repository and make it publicly available only after the paper is published. For us, these concerns are important.

Round  2

Reviewer 3 Report

The modifications made to the paper and the justifications provided are still not satisfying to assess the scientific robustness of the study. It is generally not the role of a reviewer to judge wether the study is relevant in its content given the research context, but mostly of the academic editor. Most of the response done by authors justify this relevance, but does not answer the concerns formulated in major comments. Peer review is aimed at ensuring that conclusions obtained are backed-up by robust evidence, in a sense to validate the results, the same way that reproducibility should ensure validation or falsification of research after publication. In that context, the study still have major flaws which in my opinion must be corrected for it to be publishable. These include:

  -  Regarding the research question and the contributions, it is still presented in an unclear way. At the end of the introduction (L69), it is stated that the Rank-size rule is ``the most suited'' for defining the urban system. However, it is shown in the paper that the system indeed deviates strongly from this rule. Please provide at this stage a consistent research question linked to the main argument of the paper (that the Serbian urban system is unbalanced, how to show it and that more balance would be better for development) and the main contributions.

 -  L30, the term ``optimal'' should be nuanced and/or better defined. The answer done in the response letter was satisfying and should in a way be integrated in the paper.

 -  The discussion on why polycentricity is important for the development of the urban system, which spans a large part of the conclusion, should be situated much before in the paper. In the introduction would be a good fit, to better justify why the study is relevant.

-  The definition of what is an urban system, a city and the scale considered should be made more explicit in the introduction before these terms are used, using appropriate references. An urban system can be a district, a metropolitan system, a mega-city region (where cities are not well defined), a regional urban network within a country (with different definitions for networked cities), a regional urban network across countries, a national urban system, etc. Most of it is implicit until the dataset used is described, therefore some theoretical background should be provided.

-  The choice of the two indicators considered is still not well justified, while one of the major comment was rather clear on this point: if hypothetically the urban system fits perfectly a power law, i.e. that $P_i = P_0 / i^{\alpha}$ with the cities ordered in decreasing order (in an other way if $\varepsilon$ is the error term, then $Var(\varepsilon)=0$), then $I_1 = P_1 / P_2 = 2^{\alpha}$. More generally if the system is a sequence of different rank-size laws (that is the exponent $\alpha$ varies for consecutive sequences of indices), there should be a bijection between the rank-size laws and the corresponding primacy indices. In the context of the paper, both are needed precisely because the system does not follow the rank size law, and its ``shape'' can be better accounted for by using the two complementary indices. This is not explicit enough in the current version of the paper.

-  L283 and figure 1, please provide the adjusted R-squared values and some confidence intervals for the estimated parameters. The adjustment looks visually not very good, and the difference between the hierarchy levels discussed L287 is very small and may be not significant. Proceeding to a power-law with cutoff fit following (Clauset et al., 2009) would surely yield better results; if it is not possible to do so for some reasons, at least mention it along to the linear fit performance values.

 -  The link between the historical discussion and the empirical results is still not smooth, but it may be due to the fact of having a theoretical and literature discussion after an empirical section. A solution may be to put this discussion in a dedicated section (like ``historical context'') before the methods section, or in the discussion.

 -  The term ``sustainability'' (L569) is not defined - as it is the only place it is used it may be better removed or replaced by one of the concepts used throughout the paper.

 -  Regarding reproducibility, it can only improve the research by publicly making available source code and data. The data used here has no privacy or confidentiality issue. If there is a concern on citation credits for example, preprints are a solution and also enhance the global quality of research. Anyway these are not scientific content reflexions but rather scientific policy/ethics concerns, which enforcement also belong in that case to the editorial office.

Author Response

Response to Reviewer

Thank you for your suggestions. We have made corrections in the text based on your suggestions (red text) and we hereby provide additional explanations.

Comment:

Regarding the research question and the contributions, it is still presented in an unclear way. At the end of the introduction (L69), it is stated that the Rank-size rule is ``the most suited'' for defining the urban system. However, it is shown in the paper that the system indeed deviates strongly from this rule. Please provide at this stage a consistent research question linked to the main argument of the paper (that the Serbian urban system is unbalanced, how to show it and that more balance would be better for development) and the main contributions.

Response:

The introduction has been rewritten as to avoid imprecise formulations and explain in greater detail the research question.

Comment:

L30, the term ``optimal'' should be nuanced and/or better defined. The answer done in the response letter was satisfying and should in a way be integrated in the paper.

Response:

The tern ‘optimal’ has been removed from the text and replaced with ‘balanced’. We believe the confusion is resolved in the rewritten introduction.

Comment:

The discussion on why polycentricity is important for the development of the urban system, which spans a large part of the conclusion, should be situated much before in the paper. In the introduction would be a good fit, to better justify why the study is relevant.

Response:

The discussion on polycentricity has been embedded in the rewritten introduction. Lines 94(104)-117.

Comment:

The definition of what is an urban system, a city and the scale considered should be made more explicit in the introduction before these terms are used, using appropriate references. An urban system can be a district, a metropolitan system, a mega-city region (where cities are not well defined), a regional urban network within a country (with different definitions for networked cities), a regional urban network across countries, a national urban system, etc. Most of it is implicit until the dataset used is described, therefore some theoretical background should be provided.

Response:

The definition of the urban system has been provided in a footnote and it has been made clear in the introduction that the paper deals with a national urban system.

Comment:

-  The choice of the two indicators considered is still not well justified, while one of the major comment was rather clear on this point: if hypothetically the urban system fits perfectly a power law, i.e. that $P_i = P_0 / i^{\alpha}$ with the cities ordered in decreasing order (in an other way if $\varepsilon$ is the error term, then $Var(\varepsilon)=0$), then $I_1 = P_1 / P_2 = 2^{\alpha}$. More generally if the system is a sequence of different rank-size laws (that is the exponent $\alpha$ varies for consecutive sequences of indices), there should be a bijection between the rank-size laws and the corresponding primacy indices. In the context of the paper, both are needed precisely because the system does not follow the rank size law, and its ``shape'' can be better accounted for by using the two complementary indices. This is not explicit enough in the current version of the paper.

Response

If we have correctly understood your request, as the formula is illegible, we may say that it goes beyond the scope of our research. We are aware that there are other models that may be applied and we tested some of them in the preliminary stage of research. However, finding the best model was not our aim at this stage. We sought to demonstrate that the model was not linear and that the deviation from the linear model was extreme. The purpose of the discussion was to explain the extreme deviation as the hallmark of the national urban system. Other statistical models would certainly provide additional information about particular segments of the urban system but they do not compromise the identified disproportion in urban sizes. In our opinion, including them would not significantly contribute to the study and would merely blur the focus on its main purpose.

Comment:

L283 and figure 1, please provide the adjusted R-squared values and some confidence intervals for the estimated parameters. The adjustment looks visually not very good, and the difference between the hierarchy levels discussed L287 is very small and may be not significant. Proceeding to a power-law with cutoff fit following (Clauset et al., 2009) would surely yield better results; if it is not possible to do so for some reasons, at least mention it along to the linear fit performance values.

Response:

It was not our aim to find the best model for our data but merely to demonstrate that the model is not linear. Due to this, we did not find it necessary provide the adjusted R-squared values, as it is apparent from Figure 1 that the correlation is not linear. If you consider it necessary to provide additional statistical evidence that the correlation is not linear, we can do this.

Comment:

The link between the historical discussion and the empirical results is still not smooth, but it may be due to the fact of having a theoretical and literature discussion after an empirical section. A solution may be to put this discussion in a dedicated section (like ``historical context'') before the methods section, or in the discussion.

Response:

A correction has been made in the text: the section dealing with the historical context has been placed between Methods and Results.

Comment:

The term ``sustainability'' (L569) is not defined - as it is the only place it is used it may be better removed or replaced by one of the concepts used throughout the paper.

Response:

The concept of sustainability has been introduced earlier in the text and a conceptual link between urban size, regional development and sustainable development has been established

Comment:

Regarding reproducibility, it can only improve the research by publicly making available source code and data. The data used here has no privacy or confidentiality issue. If there is a concern on citation credits for example, preprints are a solution and also enhance the global quality of research. Anyway these are not scientific content reflexions but rather scientific policy/ethics concerns, which enforcement also belong in that case to the editorial office.

Response:

The paper relies on the data that are already publicly available in the publications of the Statistical Office of the Republic of Serbia. We have added relevant citations in the list of references.

Round  3

Reviewer 3 Report

The modifications and additions made to the text make it much clearer. The research question/goals of the paper and how results answer to it are better articulated, and from this viewpoint the paper reaches a good standard. I would still suggest that more robust statistical analysis would be necessary (r squared, confidence intervals) but reporting standards highly depend on disciplines, so I would let the editor judge the necessity of these additional numerical details (but I agree that they would not bring much to the general purpose and results of the paper). Beside that, the paper can basically be accepted in its current state now.

This manuscript is a resubmission of an earlier submission. The following is a list of the peer review reports and author responses from that submission.

Round  1

Reviewer 1 Report

The paper is interesting and adds a new perspective to the current state of research in the field of urban system in Serbia. The analysis remains relevant and results are interesting. The literature review is very satisfying. It is well studied and comprehensive. The presented text meets the formal criteria of scientific studies: the empirical analysis is based on a theory and the findings are based on the conducted analysis, but in my opinion it needs a little effort before acceptance:

My main reservations are related with the presentation of data analyses methods used. The methods chosen by the authors are quite simple and are based on generally known formulas. Therefore, I have doubts whether it is necessary to present generally known formulas on the mean value (formula 6) or standard deviation (formula 7). For this reason, although the article is interesting and well-written, one should consider changing the way of describing the results.

I suggest also to explain the following symbols G1, G2 and others in text (line 219-220 and 225).

I also have doubts about the authors' conclusion on the significance of the correlation coefficient (line 318, formula 8). Is the value of correlation coefficient should be interpreted as insignificant? I think that the authors did not test the significance of the correlation coefficient. In my opinion, one should leave information that the correlation exists, but it is weak.

Reviewer 2 Report

The paper discusses the unevenness of Serbia’s urban development. The primary conclusion is the domination of Belgrade. There are a very small number of large cities with more than 100,000 inhabitants.

While I enjoyed reading the paper and the paper was informative on the evolution of population concentration of Serbian cities since 1948, I am still doubtful of the utility of the paper without an empirical and formal analysis of the uneven development. The authors do provide some discussion on this in section 4.2. But that is not sufficient. There is a large literature in urban economics on how cities develop. The authors could refer to the papers by Edward Glaeser. 

In my view, the paper is incomplete without a formal economic and empirical analysis of why the Serbian urban system became uneven.